# Interplay between Partial EMT and Cisplatin Resistance as the Drivers for Recurrence in HNSCC

**DOI:** 10.3390/biomedicines10102482

**Published:** 2022-10-05

**Authors:** Julia Ingruber, Jozsef Dudas, Susanne Sprung, Bianca Lungu, Felicitas Mungenast

**Affiliations:** 1Department of Otorhinolaryngology and Head and Neck Surgery, Medical University of Innsbruck, Austria and University Hospital of Tyrol, 6020 Innsbruck, Austria; 2Inpath GmbH, 6020 Innsbruck, Austria; 3TissueGnostics SRL, 700028 Iasi, Romania; 4TissueGnostics GmbH, 1020 Vienna, Austria

**Keywords:** partial EMT, cisplatin resistance, StrataQuest, SCC-25 cells

## Abstract

This study aims to investigate the role of partial epithelial to mesenchymal transition (pEMT)-related proteins in modulating Cisplatin resistance in head and neck squamous cell carcinoma (HNSCC). SCC-25 cells were pre-treated with TGF-beta1 followed by transient Krüppel-like Factor 4 (KLF4)-overexpression and Cisplatin treatment. Cell growth, cell morphological changes and cell migration were assessed using Juli BR live cell video-microscopy. In addition, Ki-67 and Slug immunostaining and follow-up image cytometric analysis of primary and recurrent HNSCC tumors were performed to evaluate the proliferation index (PI) and the EMT-like phenotype. We observed that proliferating and Slug-positive tumor cells expand after therapy in HNSCC. Subsequently, protein analysis revealed the stabilization of Slug, upregulation of Vimentin and phospho-p38 (p-p38) in Cisplatin-resistant SCC-25 cells. Moreover, KLF4-overexpression contributed to Cisplatin sensitivity by reduction of Slug at the protein level. This work strongly suggests that an pEMT-like pathway is activated in recurrent and Cisplatin-resistant HNSCC. Finally, stable KLF4-overexpression might sensitize HNSCC tumor cells for Cisplatin treatment.

## 1. Introduction

EMT is a reversible and complex phenotypic switch between the epithelial and mesenchymal cell type that has gained an essential role in cancer research. The most important clinical significance of EMT is its relationship with cancer cell migration, metastasis and resistance to apoptotic stimuli [1,2,3]. Loss of cellular adherens junctions of epithelial tumor cells allow them to undergo EMT, to migrate through the extracellular matrix and to disseminate into the blood circulation. Tumor-microenvironment factors such as interleukin 6 (IL-6) and TGF-beta1 promote the EMT program in progressed head and neck squamous cell carcinoma (HNSCC) by targeting and activating the expression of a variety of master EMT-transcription factors (EMT-TFs) in vivo and in real patient pathologies [3,4,5].

The EMT typical for HNSCC tumor cells is regulated by the activation of the pro EMT-TF Slug, a member of the SNAI gene family [3,6,7]. Moreover, detection of Slug, as well as reduction of E-cadherin assessed by immunohistochemical staining of cancer cell nests are statistically correlated with poor response to radiochemotherapy (R(C)T) and reduced patient overall survival in HNSCC [7,8]. During TGF-beta1-induced EMT, in SCC-25 HNSCC cells, Krüppel-like Factor 4 (KLF4) protein levels significantly decreased together with E-cadherin and HSP-70, whereas N-cadherin, Slug and Vimentin increased [4,9]. This downregulation of epithelial markers and simultaneous upregulation of Slug and mesenchymal markers characterize the classical molecular switch from an epithelial- forward to a mesenchymal phenotype. Biomechanistically, the EMT phase in tumor cells is engaged with increased cell mobility, matrix degradation activities, and reduced expression levels of the cell proliferation marker Ki-67 [10].

The EMT molecular program is plastic [11]; it can be reversed and thereby return to the epithelial phenotype, named mesenchymal to epithelial transition (MET). The reversible phenotypic switch from an epithelial to a mesenchymal phenotype and vice versa has been associated as a crucial step in embryonic development. Hereby, embryonic cells follow the EMT program to increase their migratory potential and subsequently undertake the reverse process, MET, to differentiate into various cell types [12,13]. Likewise, both directions of trans-differentiation are reactivated in tumorigenesis, however a full switch from an epithelial starting point into a differentiated mesenchymal phenotype (complete EMT) is rare to achieve. More likely is the partial EMT (pEMT) phenotype, a plastic transient state, which generates epithelial-mesenchymal tumor cells, characterized by the co-expression of specific epithelial (e.g., Cytokeratin) and mesenchymal markers (e.g., Vimentin) [7,14,15,16].

EMT cells are localized at the border of the primary tumor cell nest near to cancer-associated fibroblasts (CAFs) in the surrounding tumor microenvironment (TME) and in proximity to the TGF-beta1 source, available from blood vessels, CAFs and myofibroblasts [4,17]. Moreover, a high pEMT score was found to be associated with an increased number of lymph node metastasis and a higher tumor grade [16].

The MET-phase is described as the process whereby mesenchymal-like cells transdifferentiate to a polarized, proliferating epithelial-like cell type. Additionally, MET programs are characterized by decreased expression of mesenchymal proteins and gene products, such as Vimentin, N-cadherin and Fibronectin, accompanied by the upregulated of epithelial Cytokeratin and KLF4. In addition to this, E-cadherin alone was discussed as an important MET inducer [18].

The induction of MET is strongly influenced by TME factors [5], such as inflammation, hypoxia and oxidative stress, but also the wounds by operative therapeutic treatment might trigger the expansion of sleeping EMT cells [19]. Previous research groups in concert with our experimental data revealed that MET is associated with the (re)-activation of KLF4 [9,20] and a proliferative cell growth of R(C)T-treated cancer cells.

Moreover, we already observed an inverse correlation between KLF4 and E-cadherin on one side, and Slug on the other side, in the regulation of the process of EMT and MET in HNSCC [9]. In contrast to Slug, which is related with EMT and therapy resistance [7,8], experimental overexpression of KLF4 induced a re-epithelialization of EMT tumor cells [9]. At the same time, SNAI2 (Slug) is the only EMT-related transcription factor whose gene expression level characterizes EMT in HNSCC, based on patients’ RNA cohort and data investigated from The Human Genome Atlas (TCGA) gene expression data bank [16]. Interestingly, Tai et al. [21], reported in 2011 that persistent KLF4 expression independently correlated with worse disease-specific survival, especially in patients with advanced HNSCC. Based on this knowledge, we assessed that in progressed HNSCC both Slug and KLF4 are present simultaneously. The secondary growing epithelial cancer cell nests could retain properties from the EMT-phase, which are heterogeneous and successfully protect themselves against R(C)T.

The main hypothesis of this work is: HNSCC tumor cells are in pEMT and represent expression of both epithelial and mesenchymal phenotypes. Upon environmental stimulation mesenchymal characteristics are increased, cells represent EMT, cell growth reduces combined with therapy resistance. When the exposition of the stimuli is ended, HNSCC tumor cells return to epithelial proliferation, gain MET characteristics, but they still represent the pEMT-related therapy resistance. MET cells are not identical with the original epithelial tumor cells; although they are a proliferating cell type expressing epithelial markers and representing epithelial morphology, they are still not vulnerable to Cisplatin treatment at clinically relevant doses (10 µM) [22]. We further assume that for the pEMT-related therapy failure, the non-canonical TGF-beta1-signalling pathway via p38 MAPK is responsible. Subsequently, we test the hypothesis that the process of EMT/MET switch enables the expansion of therapy-resistant tumor cells in HNSCC.

As a further hypothesis, we assume that environmental factor TGF-beta1 transiently reduces cancer cell proliferation and triggers the switch of the epithelial to mesenchymal phenotype in stressed conditions of the cancer cells in vivo. According to our own results, TGF-beta1 induces the phosphorylation and activation of p38 mitogen-activated protein kinase (MAPK) in HNSCC cell lines [4], which seems to be engaged with an increased expression of EMT-TF Slug. Wu et.al described in 2016 this noncanonical TGF-beta1-signaling pathway via p38 MAPK as a possible alternative to the canonical TGF-beta1-signaling pathway via Smad [23].

## 2. Material and Methods

### 2.1. Patients’ Profile

In this study, patients’ biopsies with incident head and neck squamous cell carcinoma (HNSCC) treated between 2013 and 2021 at the Department of Otorhinolaryngology-Head and Neck Surgery, Medical University of Innsbruck were included for immunohistochemical investigation and TissueFAXS analysis (TissueGnostics, Vienna, Austria). Tumor tissue samples used in this study were surgically removed during pan-endoscopy following patients’ written informed consent, approved by our Ethics Committee of the Medical University of Innsbruck (Approval ID: UN4428, ethic commission meeting of 303/4,14, 26 July 2011). All relevant clinical data were stored in a clinical tumor registry, where actually (May 2022) 2424 patients are included. For further analysis, patients with primary, recurrence/residual and secondary tumor types were selected based on availability of archived formalin fixed and paraffin embedded (FFPE) tumor tissue samples. The study population is summarized in Appendix A.

### 2.2. Enzyme Immunohistochemistry and TissueFAXS Acquisition

HNSCC tumor tissue samples from patients were used for enzyme immunohistochemical detection (Ventana, Discovery Immunostainer, Tucson, AZ, USA) of Slug, KLF4, and Ki-67 primary antibodies [24]. Antibodies were used and diluted as suggested by the manufacturers (further details are listed in Appendix A) All used tumor tissue samples were paraffin-embedded and sectioned, as published previously [25]. Immunohistochemistry was completed using a universal secondary antibody (Roche Ventana, Oro Valley, AZ, USA) and the Ventana DAB Map kit. Slug, KLF4 and Ki-67 protein expression was deemed positive when the nuclei of the tumor cells showed a specific brown staining reaction. The specific immunohistochemistry labelling levels were acquired and visualised in TissueFAXS brightfield tissue cytometer using a Pixelink camera (Pixelink, Rochester, NY, USA). Image analysis was conducted with the single cell analysis software HistoQuest 7.143 (TissueGnostics, Vienna, Austria) [4,9]. Haematoxylin was used for single-cell identification. The percentage ratio of positive cells in primary and progressed HNSCC tumor tissue samples was evaluated as described earlier [9].

### 2.3. Ki-67 Level Quantification by StrataQuest in Primary Tumor and Recurrence Tumor Slides

Acquired Ki-67-stained whole slide images were imported in the form of TissueFAXS files into the contextual image analysis software StrataQuest 7.0 (TissueGnostics, Vienna, Austria), and whole regions or regions of interest were chosen for investigation of Ki-67-positive-cells in cancer cell nests and in stroma area (Appendix A). A specially developed StrataQuest application of TissueGnostics was used to automatically stratify the tissue in macrostructures (cancer cell nests, epithelial and stroma areas and background) and to determine Ki-67 positive cells (those were determined as mixed signal of Haematoxylin and DAB using the “Single Reference Shade” algorithm). The background, cancer cell nests, epithelial and stroma areas are distinguished by the StrataQuest machine-learning-based ‘Classifier engine’, which identifies to which of a set of categories (classes) a new observation (pixel or set of pixels) belongs, based on a training set of data containing observations whose category membership is known. The individual observations are analysed into a set of quantifiable properties, known as features. In case of not-satisfactory recognition of the classified areas based in tissue heterogeneity, it was possible to manually label those and using the learning function of the algorithm to teach it accordingly. The final macrostructures were identified based on some post-processing steps: the removal of small areas/larger areas and if necessary manual correction using manual annotations provided by the user. The cancer cell nests and the epithelium were separated based on their area and intensity, and the separation succeeded by thresholding, which could be adjusted by the end user. The output of the tissue classification is visualized in Appendix A. Using the tumor and stroma masks, obtained with the classifier engine, the cells were grouped into two categories, tumor cells vs. stroma cells, as shown in Appendix A. Single cells were recognized by the haematoxylin staining of the cell nuclei and the Ki-67 signal was determined within the nuclei mask. A cut-off on Ki-67 intensity within a scatterplot was set manually to determine which cells are Ki-67 positive or negative. As a verification step the cells identified as Ki-67-positive were evaluated by the backward connection function, a process by which detected objects are linked to their corresponding position in a scatterplot of StrataQuest. The following determined gates were used for final data acquisition from the application: “nuclei in tumor”, “nuclei in stroma”, “tumor cells embedded in stroma”. The results are visualized in Appendix A. These event groups were determined based on a set of morphological measurements like nuclei area and compactness and also by intensity measurements (mean intensity).

For final results the end user determined the percent of Ki-67-positive (proliferating) and negative (not proliferating) tumor cells in cancer cell nests, as well as Ki-67-positive (proliferating) and negative (not proliferating) tumor cells in stroma (Appendix A). The number of proliferating tumor cells related to the number of proliferating and not proliferating tumor cells together in gave the proliferation index (PI), which was used as the final result. The application and its utilization were developed by the supervision of an experienced head and neck pathologist (SS).

### 2.4. Cell Culture, Cell Transfection and Generation of an EMT/MET Model by Ectopic KLF4-Overexpression and TGF-Beta1 Treatment

SCC-25 and UPCI-SCC090 cells were used in the experiments. Both of them were purchased from the German Collection of Microorganisms and Cell Cultures (DSMZ, Braunschweig, Germany). SCC-25 cells were cultured in DMEM/Ham’s F12 medium, UPCI-SCC090 cells in EMEM medium supplemented with 10% FBS, 2 mM l-glutamine, 100 units/mL penicillin and 100 μg/mL streptomycin (All from Pan-Biotech, Aidenbach, Germany).

Thirty thousand cells per ml were plated on six-well-plates. After 24 h, all cells were supplied with serum-free DMEM-F12 medium (serum proteins were replaced with bovine serum albumin (Serva, Heidelberg, Germany)). From day 2 until day 5 cells were treated with serum-free medium optionally supplemented with 1 ng/mL TGF-beta1 (RnD Systems, Minneapolis, MN, USA). From day 5 until day 8 cells were treated with serum-free medium optionally supplemented with 1 ng/mL TGF-beta1, or with 10 µM Cisplatin (Selleckchem, Houston, TX, USA) or with combined. All experiments were repeated as complete three biological repeats. After the treatments were completed, the cells were scraped into RIPA buffer [4,9] and used for protein isolation and Western blot [4,9].

Based on the results of the simultaneous TGF-beta1 and Cisplatin treatment a further sequential treatment schedule was developed. Seven thousand SCC-25 cells per ml were plated on six-well-plates. After 24 h, all cells were supplied with serum-free DMEM-F12 medium (serum proteins were replaced with bovine serum albumin). From day 2 until day 6 and from day 6 until day 8 cells were treated with serum-free medium or with 1 ng/mL TGF-beta1, which latter resulted in cell growth arrest. On days 8 and 9 cells were transfected with 2 µg/well pCMV3-controlled expression vectors (empty vector and KLF4-overexpressing vector, Sinobiological, Beijing, China) mixed with 4 µL ViaFECT (Promega, Madison, WI, USA) in 200 µL Opti-MEM (Gibco, Darmstadt, Germany) and further cultured from day 10 until 13 with 10% serum-containing DMEM-F12 medium. On day 13 cells were 24 h treated with 10 µM Cisplatin, passaged and replated at 3 × 10^4^ cells/mL. Cells were cultured for 9 days and used for protein analysis.

Cell growth was evaluated by confluency measurements using the Juli BR live cell video microscope (Nanoentek, Seoul, South Korea) 2, 4, 7 and 9 days after replating following the Cisplatin treatment. Live video acquisition was done for 72 h at 15 min frames during TGF-beta1-induced EMT, after TGF-beta1 treatment in MET-phase and after Cisplatin treatment. In addition, phase contrast images were taken using the Juli BR microscope.

After completion of treatments (9 days after replating following the Cisplatin treatment), cells were used for protein analysis and scraped into RIPA buffer, as published by us before [4,9].

### 2.5. Western Blot Analysis to Detect the Different Protein Levels of EMT/MET Markers in SCC-25 Cells

Western blot detection was performed to measure the protein expression of Slug, Vimentin, Phosphor-p38 MAPK, E-Cadherin and KLF4 in original and transfected SCC-25 cells treated with TGF-beta1 and/or Cisplatin. Cells were lysed on ice by 1 mL RIPA lysis buffer (50 mM Tris HCl/pH: 7.4, 1 mM EDTA, 0.5 mM EGTA, 1% Triton X-100, 0.25% sodium deoxycholate 0.1% sodium dodecylsulfate, 150 mM NaCl, 10 mM NaF, 1 mM PMSF (all reagents from Sigma, Darmstadt, Germany, except PMSF from Serva) supplemented with 10 µL HALT proteinase inhibitor cocktail (Invitrogen, Darmstadt, Germany) [4,9]. The clear supernatant was collected, and the total protein concentrations were determined using the Pierce 660 nm protein assay (Pierce, Rochford, IL, USA), according to the guidelines of the manufacturer. 10 µg protein from all samples per well was loaded on Invitrogen NuPage gels for western blotting. Nitrocellulose membranes were blocked with 5 mL (TBS) blocking buffer (Starting Block; Invitrogen-Thermo Scientific, Cat.Nr. 37542) for 1 h. Afterwards, the membranes were incubated with the specific primary antibodies overnight at 4 °C and then incubated with the secondary antibodies for 1 h at room temperature. The secondary antibodies were conjugated with horseradish peroxidase or with near infrared fluorescent markers. The peroxidase reactions were visualized with Radiance Plus substrate kit from Azure Biosytems (Sierra Ct, Dublin, CA, USA). The whole blotting procedure was handled as published elsewhere [4,26], using specific primary antibodies: rabbit monoclonal anti-KLF4, mouse monoclonal anti-E-cadherin, rabbit monoclonal anti-Slug, rabbit monoclonal phospho-p38 MAPK and rabbit monoclonal anti-Vimentin (SP-20). The completed blots were acquired in an Azure C500 documentation system. All antibodies were diluted and used according to the manufacturer protocols (further details are listed in Appendix A). For signal detection, near infrared fluorescence conjugated labelled secondary antibodies (anti-rabbit IgG and anti-mouse IgG) were used, both available from Li-Cor Biosciences (Bad Homburg, Germany) and Azure Biosystems (Appendix A). Low abundant proteins as anti-phospho-p38 MAPK and anti-Slug were visualized using chemiluminescence detection system [4,9] and peroxidase labelled anti-mouse or anti-rabbit secondary antibodies (Appendix A). Optical densities of the proteins of interest and GAPDH were measured in Image Studio Lite, version 5.2 (Li-cor). GAPDH was detected in the same membrane as the proteins of interest and allowed the loading control normalization of the samples [4,9]. The mean of the normalized optical densities of the proteins of interest in the controls were set to “1” and all treated samples were related to the control.

### 2.6. Statistical Analysis

All experiments were repeated three times. The quantification of Western blots is based on six Western blots. Cell growth data and Western blot optical densities data sets were tested for normal distribution using the D’Agostino and Pearson omnibus normality test.

Comparison of two data sets was done with Student´s t-test in case of normal distributed data, or with Wilcoxon matched-pairs signed rank test in case of not parametric data. Western blot samples were compared in pairwise analysis. Comparisons of more data sets were conducted by parametric ANOVA or non-parametric Kruskal–Wallis test.

Statistical analysis was performed by SPSS Version 27 (IBM, Chicago, IL, USA) and by Graphpad Prism Version 9 (San Diego, CA, USA).

## 3. Results

### 3.1. Slug-Positive Cells Expand after Therapy in HNSCC

Slug is a strong predictive factor for R(C)T treatment failure and supports decisions for operative first therapy whenever possible in HNSCC [8]. In 2021, we reported about a negative relationship between the gene expression of the pro EMT-TF Slug and the anti EMT-TF KLF4, analysed in HNSCC patient tissues [4]. In addition, in HPV-positive carcinogenesis the KLF4/Slug switch was not observed, and tumor cells do not develop the pEMT-like and Slug-positive phenotype.

To evaluate the roles of Slug and KLF4 in the two subgroups (1) primary tumor (n = 58) and (2) recurrence or second tumor (n = 14), immunohistochemistry and TissueFAXS image cytometric analysis were performed. The data revealed that the proportion of Slug-positive cells was significantly higher in recurrence/second tumor samples (*p* = 0.007), whereas KLF4 was present at the same level in primary and progressed HNSCC (recurrence or second tumor) (Figure 1A,B). The samples were not normally distributed. Mann–Whitney U-test was used to investigate the central tendencies between the two defined subgroups (primary tumor; recurrence or second tumor). Additionally, Slug overexpression was visualized and pathologically interpreted immunohistochemically in primary and recurrence HPV-negative HNSCC.

The determination of the overexpression of Slug was done exactly from the same patients, (Figure 1C–H, left and right panels in the same row are identical patients) respectively before and after therapy. In primary tumors the Slug staining was limited to the edge of the cancer cell nest near the tumor–stroma interface or was not present (Figure 1C,E,G). Cases that did not respond to the first therapy showed an increased and partially diffused Slug immunohistochemical staining reaction (Figure 1D,F,H).

Next, to further investigate the proliferation level in primary tumor versus recurrence tumor, Ki-67 antibody was used to measure the growth fraction of HNSCC tumor cells in the cancer cell nest and in the tumor stroma.

### 3.2. Disseminating Proliferating Cells Spread after Therapy in HNSCC

Ki-67 expression was assessed by routine IHC on available FFPE samples. Of these, patients with primary (before therapy) and recurrence (after therapy) archived FFPE were chosen for image acquisition in TissueFAXS system and Ki-67 quantification using StrataQuest 7.0 application (Appendix A). In the sample subset all the patients were male. Whole regions or regions of interest (ROIs) were chosen to evaluate the Ki-67 proliferation index (PI, relative abundance of Ki-67-positive cells) in the cancer cell nests and in the stroma regions of primary (n = 16) and corresponding recurrence tumors (n = 24). Further, the PI of primary and recurrence tumors in case of surgery (n = 9) and R(C)T (n = 7) was analysed.

No significant difference between the PI in tumor cells in the cancer cell nests before and after therapy was observed at 95% confidence interval (Figure 2A). However, if the PI of tumor cells embedded in the stroma regions before and after therapy were compared, the proliferation rate was significantly higher in recurrence tumors than in primary tumors (*p* = 0.02) (Figure 2B). Moreover, the PI in the disseminated tumor cells was significantly increased in recurrence samples in case of surgery (*p* = 0.024, Mann–Whitney U-test) (Figure 2C). Due to the relatively low number of available regions, the visible difference in case of R(C)T was not statistically different (*p* = 0.181, with Mann–Whitney U-test). These results suggested: after therapy, especially in case of surgery, circulating and disseminated HNSCC tumor cells had a higher cell proliferation and migratory potential than before therapy.

### 3.3. pEMT Enables the Expansion of Therapy-Resistant Cells in HNSCC via Single and Collective Cell Migration

Tumor cells can migrate as individual circulating tumor cells and in whole multicellular clusters, or using combined intermediate movement types [27,28]. First, we could confirm the result of other researchers, that pEMT can be associated with a mixture of single-mesenchymal cell migration and collective cell migration.

In our actual experiments, TGF-beta1 treatment in SCC-25 cells induced a stable EMT phenotype and cell growth arrest in three independent repeats. Cell growth restarted after 20 h. Further, 16–18 h after TGF-beta1 treatment, tumor cells showing the first individual mesenchymal migration were noticed, whereas the collective cell migration was detected after 20 h. The mesenchymal cell migration was characterized by an elongated, spindle-like cell shape with the morphological formation of a fibroblastic phenotype (Figure 3A). In contrast, the collective cell migration was observed as a coordinated movement of multicellular groups with an increased invasive capacity and epithelial phenotype (Appendix A). Furthermore, the collective cell migration in the MET phase was characterized by the maintenance of two distinct cell populations: leader and follower cells (Figure 3B). Leader cells were observed at the leading edge of the tumor cell accumulation in SCC-25 cells; this population indicated the direction by movements towards other epithelial cell clusters, whereas follower cells were located in the cell reservoir, moving along with the leader cells [29].

The MET phase was accompanied by uncontrolled cell divisions and cell proliferation 36 h after TGF-beta1 treatment.

### 3.4. Interplay between TGF-Beta1-Induced EMT and Cisplatin in Simultaneous Treatment System

Slug and EMT are related with R(C)T resistance [2,3,8,17]. To further confirm this relationship in HNSCC, protein levels of Slug (Figure 4A and Appendix A), Vimentin (Figure 4B and Appendix A) p-p38 (MAPK) (Figure 4C and Appendix A) as pro EMT factors and KLF4 (Figure 4D and Appendix A) and E-cadherin (Figure 4E and Appendix A) as epithelial markers were analysed by Western blotting system in human SCC-25 (HPV^−^) and UPCI-SCC90 (HPV^+^) cell lines. The data revealed that Vimentin protein levels increased by TGF-beta1, and by combined treatments (Figure 4B and Appendix A) in the SCC25 experimental model. In this cell line Cisplatin treatment strongly induced Slug protein expression, which further increased by combined treatments (Figure 4A and Appendix A). Cisplatin also induced p38-MAPK phosphorylation (Figure 4C and Appendix A). Following TGF-beta1 treatments, we observed rapid loss of KLF4 protein levels in SCC25 cells, independently from Cisplatin co-treatment (Figure 4D and Appendix A). The cell adhesion protein E-cadherin slightly changed by TGF-beta1-induced EMT and did not respond to Cisplatin treatment (Figure 4E and Appendix A).

The same experimental procedure was performed with UPCI-SCC090 cells (Appendix A). Considering EMT-markers Vimentin was very low detected using chemiluminescence and did not show any reaction on TGF-beta1 or Cisplatin treatments. Slug or p-38 (MAPK) were not detected at all. In consequence, further experiments using EMT-induction by 6 days TGF-beta1 treatment followed by MET and transfections with KLF4-overexpressing vectors and Cisplatin treatment were done using SCC-25 cells and not with UPCI-SCC090 cells.

### 3.5. KLF4-Overexpression Downregulates Slug, Vimentin and p-p38 after TGF-Beta1-Induced EMT

Our recent evidence suggests that in a background of high and stable KLF4 protein levels, the antagonist Slug cannot be activated (HPV-positive HNSCC tissue samples, UPCI-SCC-90 cells) [4,9], and in this condition HNSCC is sensitive to Cisplatin treatment.

Western blot detection of Slug, Vimentin (Figure 5A,B and Appendix A), phospho-p38 (MAPK) (Figure 5C and Appendix A) and E-cadherin (Figure 5D and Appendix A) was performed in SCC-25 cells after TGF-beta1-pretreatment, KLF4-overexpression, and 9 days after Cisplatin treatment. Vimentin, Slug and E-Cadherin were clearly detectable using fluorescence imaging system (Appendix A). Phospho-p38 MAPK signaling was detected using highly sensitive chemiluminescence (Appendix A).

Here we observed that although 5 days had passed after TGF-beta1-pretreatment and medium replacement, a still significant ten-times-upregulation of Vimentin (Figure 5B, *p* = 0.03, with Wilcoxon test) was present (Appendix A). The normalized and relative Slug OD values were not significantly increased in TGF-beta1-pretreated samples compared to control (Figure 5A, *p* = 0.062, with Wilcoxon test), whereas the p-p38 values significantly increased to 2.5 times (*p* = 0.015, with paired *t*-test, Figure 5C).

Forced overexpression of KLF4 compared to empty vector transfection (control condition), lead to significant, 50% downregulation of Slug (Figure 5A, *p* = 0.03, with Wilcoxon test), and similarly, both Vimentin (Figure 5B, *p* = 0.03, with Wilcoxon test) and p-p38 (Figure 5C, *p* = 0.012, with paired t-test,) protein levels were significantly decreased, 40%, 60% respectively, while E-cadherin remained stable (Figure 5D, *p* = 0.21, with paired t-test). This result is an important agreement with our previously published data, that if KLF4 is stable, Slug will be reduced [4]. This reduction was accompanied by Vimentin and p-p38 to a similar extent.

Nine days of Cisplatin treatment resulted in stable constitutive Slug protein levels (Figure 5A), whereas p-p38 levels were four-times significantly upregulated (Figure 5C, *p* = 10^−4^, with paired t-test). The Vimentin protein levels also significantly increased in Cisplatin-treated and combined with TGF-beta1-pretreated or KLF4-overexpressed conditions (Figure 5B, *p* < 10^−4^, with Kruskal–Wallis test, pairwise Wilcoxon tests with the control also resulted in significant changes, Figure 5B). Various, but not significantly different E-cadherin levels were present after Cisplatin treatment in combination with (or without) TGF-beta1-pretreated or KLF4-overexpressed conditions (Figure 5D *p* = 0.28, by one-way ANOVA).

In addition to the findings on the Western blots (Figure 5), our in-silico analysis using the Ensembl databank (Cambridge, United Kingdom) revealed a potential of direct regulation of Slug by KLF4 (Appendix A).

### 3.6. Transient KLF4-Overexpression Sensitizes SCC-25 Cells to Cisplatin

The Western blot analysis showed the endpoint of the experimental schedule: TGF-beta1-pretreatment—KLF4-forced-overexpression—Cisplatin treatment. Using confluence measurements by the Nanoentek live videomicroscopic system Juli BR, we were able to follow the growth of the cells in control, transfected with control vector, KLF4-overexpressed and in TGF-beta1-pretreated conditions (Figure 6). Following TGF-beta1-pretreatment SCC-25 cells were transfected with KLF4-overexpressing-vector, followed by Cisplatin treatment. Cell growth confluence data were collected 2, 4, 7 and 9 days after 24 h of Cisplatin treatment (Figure 6A). If SCC-25 cells were transfected with KLF4-overexpressing vector, and subsequently treated with Cisplatin, 7 days after Cisplatin treatment they showed a significant growth reduction (Figure 6B, *p* = 0.005 by unpaired Student´s *t*-test). Our data revealed that SCC-25 cells grow exponentially when they are recovered from a TGF-beta1-induced EMT and cell growth arrest (Figure 6A), and 10 µM Cisplatin treatment does not inhibit this growth, although KLF4-overexpression transiently might achieve a significant growth reduction even in these (Figure 6B, *p* = 0.011, by unpaired Student’s *t*-test). This growth reduction is valid 7 days after the Cisplatin treatment, so a long time is needed until the direct DNA-damage effects of Cisplatin are converted to cell growth reduction.

As mentioned before, SCC-25 cells were 5 days TGF-beta1-pretreated, transfected with pCMV3 control vector or KLF4-overexpressing vector and afterwards treated with 10 µM Cisplatin. All morphological-phase contrast images that are presented in Figure 7, were taken 10 days after Cisplatin treatment.

Compared to the control-pCMV3-vector-transfected cells (Figure 7A) and the TGF-beta1-pretreated pCMV3-vector-transfected cells (Figure 7C), the KLF4-overexpressing-vector-transfected cells at day 7 after Cisplatin treatment showed a significant growth reduction and a heterogenetic cell morphology (Figure 7B,D). In more detail, the KLF4-transfected cells did not proliferate or proliferated very slowly, and the plates contained more vacuolized cells (Figure 7B,D) and giant cells (Figure 7B,D) with a highly enlarged nucleus or multiple nuclei. Western blot analysis confirmed that KLF4 acts as a potential MET-inducing transcription factor by suppressing the classical EMT-TF Slug in HNSCC cell line SCC-25 (Figure 5, Appendix A); furthermore, phase contrast microscopy images showed that KLF4-overexpression made originally Cisplatin-resistant HNSCC cells transiently sensitive for Cisplatin treatment in vitro.

## 4. Discussion

Cisplatin, a platinum-based chemotherapeutic drug, is used for curative treatment of primary tumor and is the first option to treat residual/recurrence and metastatic HNSCC [30,31]. Unfortunately, this treatment option often achieves success only for a short time because many patients develop resistance to Cisplatin [31,32], which is followed by poor survival. The mechanism of Cisplatin resistance is a product of multiple complex epigenetic and genetic changes at molecular and cellular levels. Many of these changes promote cell survival, including improved DNA-damage repair, drug detoxification, apoptosis inhibition and alterations in drug transport pathways [31,33,34].

Interestingly, plasticity in HNSCC and the pEMT phenotype [7,8,16] has been consistently linked to therapy failure, especially to Cisplatin resistance [1,8,31,35]. pEMT cells have the potential to repair DNA damage [36], which enables tumor cells to survive Cisplatin treatments. After clinical levels of Cisplatin treatment in vitro (10 µM in vitro Cisplatin treatment corresponds to 100 mg/m^2^ Cisplatin treatment in patients [22]), the following phases were observed in resistant HNSCC cells: (1) reduction of tumor cell growth, (2) giant cells survive Cisplatin treatment but do not proliferate, (3) addition of stimulatory factors enhance the regrowth of resistant epithelial cancer cell nests from the surviving giant cells [17,37]. Nevertheless, HNSCC reacts differently to Cisplatin chemotherapy, based on its HPV-background. In HPV-positive HNSCC wild-type p53, KLF4 [9,38] and FOXA1 [39] might ensure that Slug protein cannot be stabilized, and tumor cells do not develop to the R(C)T-resistant, EMT-like and Slug-positive phenotype. In HPV-negative HNSCC, p53 is either not present or its function is lost or changed by mutations.

The main EMT-regulating transcription factor is Slug in HNSCC [16], which is significantly upregulated in tumor recurrence (*p* = 0.007, Figure 1) compared to primary tumors. Moreover, it is likely that Slug directly mediates chemoresistance in progressed HNSCC by promoting DNA repair after Cisplatin treatment, as reported previously [31,40]. However, pEMT and chemoresistance must not act together; they are separate processes, which is indicated by the results that both, TGF-beta1- and Cisplatin treatments could increase p38 MAPK phosphorylation and Slug independently.

The originally described function of Slug was the repression of E-cadherin and induction of EMT from epithelial tumor cells [41]. Wild-type p53 [42] and KLF4 [43] ensure the stable E-cadherin expression and function. Loss of E-cadherin, which might occur through repression by Slug [41], or its replacement by mesenchymal-type cell adhesion molecules as N-cadherin [9], play an important role in tumor progression, including the pEMT process and cancer dissemination [6,44,45]. Interestingly, a stable expression of the cell adhesion protein E-cadherin was found in surviving cells after Cisplatin treatment (Figure 5D). This provides evidence that downregulation of E-cadherin is not required in HNSCC for chemotherapy resistance. Moreover, E-cadherin and mesenchymal markers as Vimentin are co-expressed in tumor cells that survive and grow after Cisplatin treatment (Figure 5A–D).

As mentioned before, Slug might be an important repressor of E-cadherin. In this context the Slug-antagonist KLF4 and its protein stabilization might be interesting. In 2021, Subbalakshmi et al. [20] highlighted KLF4 as a potential MET-inducing transcription factor by suppressing classical EMT-TFs including Slug. In this current study we confirmed the inverse relationship of Slug and KLF4, where transient transfection of SCC-25 cells after EMT with pCMV3-KLF4 expression vector suppressed Slug protein levels. Moreover, KLF4-overexpression significantly reduced p38 MAPK signalling 96 h after transfection, and allowed improved Cisplatin growth inhibitory effects, which is in line with previous reports [46,47]. As described by Chen et al. [48], KLF4 increased the sensitivity of Cisplatin by the induction of apoptosis and cell cycle arrest in the S-phase in esophageal squamous cell carcinoma (ESCC) cells.

As we previously published, the Slug activation is in close context with the non-canonical TGF-beta1-signalling over the p38 mitogen-activated protein kinase (MAPK) pathway, which downregulated KLF4 and increased Slug and Vimentin at protein level [4]. Our novel data evidenced that in Cisplatin pre-treated SCC-25 cells the pEMT phenotype was still present, which was exemplified by increased mesenchymal Vimentin, stabilized Slug levels and the phosphorylation of p38 (Figure 5). Further, IHC analysis of recurrence vs. primary tumors using StrataQuest application revealed that a pEMT-like phenotype is activated in recurrence of R(C)T-resistant HNSCC, suggesting that this phenomenon may not be limited to in vitro effects.

The dual role of p38 MAPK in tumorigeneses was recently described by Martinez-Limon et al., in 2020 [49], which reflects its anti-tumorigenic activity related to cell cycle arrest, apoptosis and preventing cell proliferation and its tumor-promoting role, characterized by tumor cell migration, resistance to apoptotic stress, chemotherapeutic drugs [50] and DNA damage repair. In addition, if p38 was inhibited, resistant HNSCC cells showed a higher DNA damage response [51] and exhibited a decreased expression of cancer stem-cell markers such as CD44 and KLF4 [51,52].

As reported by our clinic [8], Slug positively correlated with the scores for the Ki-67 proliferation index (PI) in HNSCC tissue samples. Here we found that the PI of disseminated tumor cells in the stroma was significantly higher in recurrence tumors compared to primary tumors (*p* = 0.004, Figure 2). The study was limited by the availability of patient’s tissue before and after therapy.

In general, our findings suggest that proteins known to regulate the pEMT and especially the p38MAPK pathway directly contribute to Cisplatin resistance in our HNSCC model cell line SCC-25. Through transient overexpression of KLF4 we were able to inhibit the most important EMT-TF Slug in HNSCC at protein level and subsequently shut down the EMT process for a short time. The manipulation of Slug by KLF4-overexpressing vector inhibited cell growth and re-sensitized originally drug-resistant cells to the effects of Cisplatin treatment.

In HPV-positive HNSCC, the KLF4 is stabilized by a miRNA pattern [38], which allows a more efficient response to R(C)T. The investigation of this relationship is the main direction of our next research work.

## Figures and Tables

**Figure 1 biomedicines-10-02482-f001:**
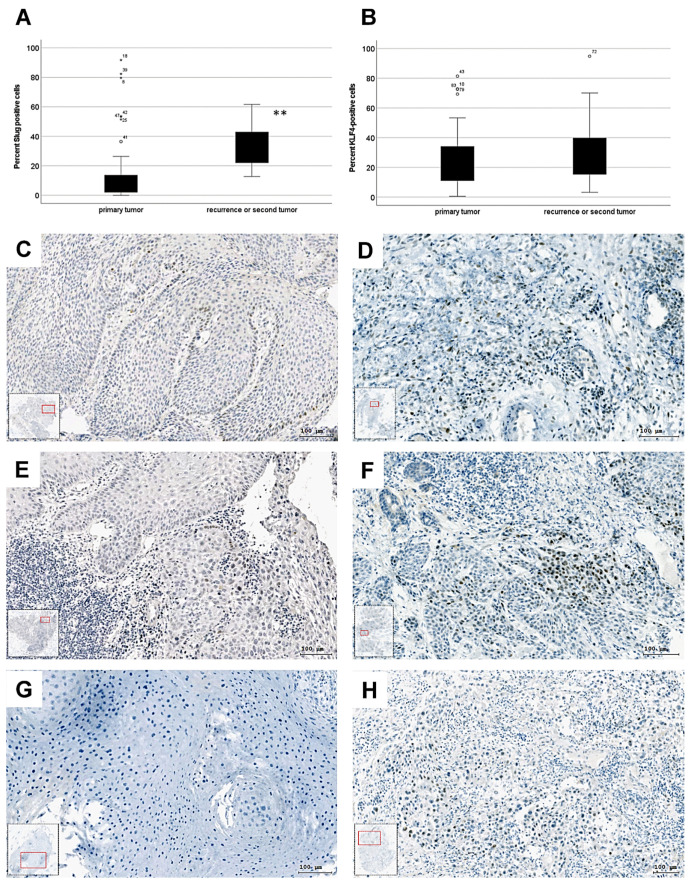
Boxplot presentation of the percentage of (**A**) Slug-positive (**B**) KLF4-positive cells with immunohistochemistry in 58 primary tumor and 14 recurrence or secondary tumor samples, (**: *p* = 0.007 with Mann–Whitney U test, KLF4-graph; *p* = 0.5). **: less than 0.01. (**C**) Slug^+^ immunohistochemical reaction in the leading edge of the cancer cell nests in HPV-negative primary oropharynx squamous cell carcinoma. (**D**) Diffuse and increased Slug^+^ immunohistochemical reaction in the recurrence of HPV-negative primary oropharynx squamous cell carcinoma after R(C)T primary treatment. (**E**) Scattered Slug^+^ immunohistochemical reaction in the leading edge of the cancer cell nests in HPV-negative primary oropharynx squamous cell carcinoma. (**F**) Diffuse and increased Slug^+^ immunohistochemical reaction in the recurrence of HPV-negative primary oropharynx squamous cell carcinoma after surgical and PORT primary treatment. (**G**) Slug^−^ immunohistochemical reaction in HPV-negative primary larynx squamous cell carcinoma. (**H**) Diffuse Slug^+^ immunohistochemical reaction in the recurrence of HPV-negative primary larynx squamous cell carcinoma after surgical and PORT primary treatment. (**C**–**H**) left and right panels in the same row are identical patients).

**Figure 2 biomedicines-10-02482-f002:**
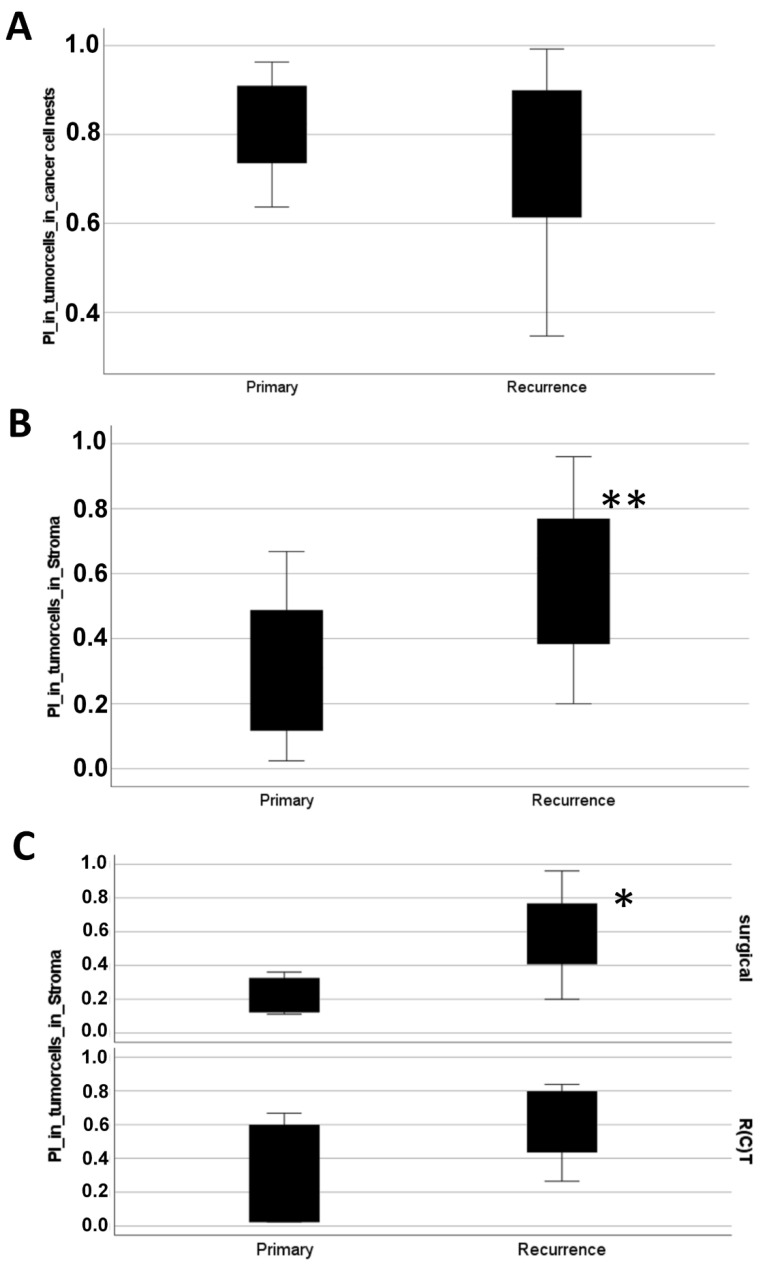
Mann–Whitney U-test was performed to compare the proliferation index (PI, relative abundance of Ki-67-positive cells) of tumor cells in cancer cell nests (**A**) and tumor cells found in stroma areas (**B**). (**A**) For comparison of the PI in tumor cells in cancer cell nests 16 regions of therapy naive primary tumors and 24 regions of recurrence tumors were available. No significant difference in the PI in the tumor cell nests was found (*p* = 0.576, at 95% confidence interval n = 40). (**B**) The PI in tumor cells embedded in stroma (disseminating tumor cells) was significantly higher in recurrence or residual tumors (20 regions were available) than in therapy naive primary tumors (12 regions were available), (*p* = 0.004; at 95% confidence interval). (**C**) Comparison of primary and recurrence tumors in case of surgery and R(C)T with a significant higher PI in disseminated stromal tumor cells in case of surgery treatment (*p* = 0.024, n = 9). *: less than 0.05; **: less than 0.01.

**Figure 3 biomedicines-10-02482-f003:**
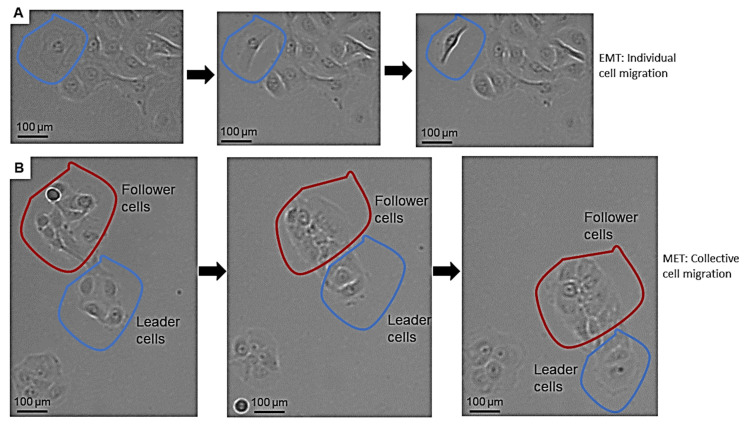
Distribution of tumor cells in pEMT. In pEMT tumor cells might detach as individual cells and migrate away from the cancer cell nest (**A**), or a whole cancer cell group with epithelial morphology might migrate 30–40 µm per hour (**B**) by collective cell migration together.

**Figure 4 biomedicines-10-02482-f004:**
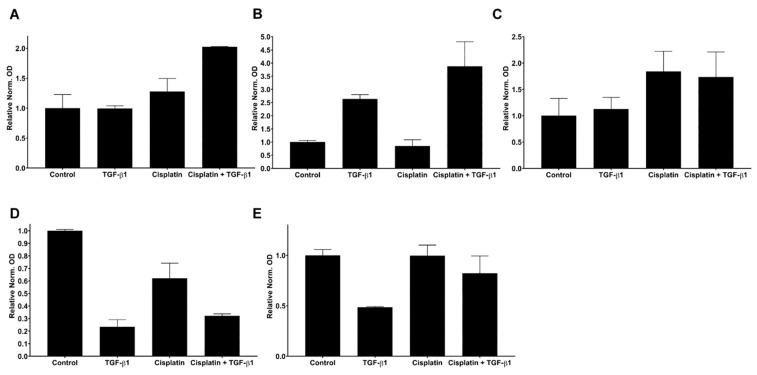
Quantification of the protein levels of Slug (**A**), Vimentin (**B**), p-p38 (MAPK) (**C**), KLF4 (**D**) and E-cadherin (**E**) from western blots presented in Appendix A. GAPDH was used as loading control. All experiments were repeated as complete three biological repeats. Optical densities (ODs) of the bands of proteins of interest were normalized with the optical densities of GAPDH. The mean of the normalized ODs of control samples (only serum-free culture condition) were set to “1”.

**Figure 5 biomedicines-10-02482-f005:**
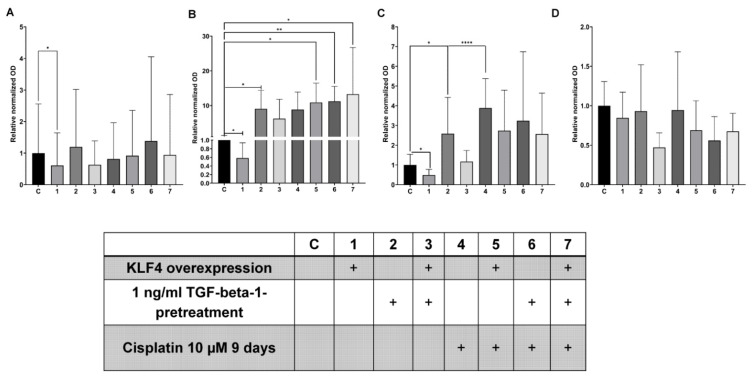
Quantification of the optical densities (ODs) of bands of six membranes for Slug (**A**), Vimentin (**B**), p-p38 (**C**) and E-cadherin (**D**). The optical densities were normalized with the corresponding GAPDH bands, and the mean value of the control (“C” in the explanatory table) was set to 1. The normalized OD of the treated samples was related to the mean value of the controls (from six repeats). The relative ODs for the treatments listed in segment lower panel (treatment table) are presented on the graphs (**A**–**D**). KLF4-overexpression resulted in 40–60% decrease of Slug, Vimentin, p-p38. Slug protein levels remained constitutive after TGF-beta1 and Cisplatin treatments. Both Vimentin and p-p38 levels increased substantially after TGF-beta1 and Cisplatin treatments. *: less than 0.05; **: less than 0.01; ****: less than 0.0001.

**Figure 6 biomedicines-10-02482-f006:**
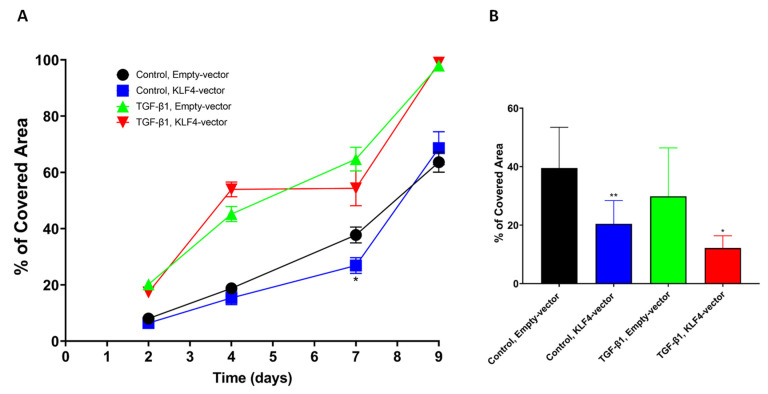
Transient pCMV3-regulated overexpression of KLF4 allows a transient growth reduction of SCC-25 cells (**A**). Twenty-four hours of Cisplatin treatment and plating of 3 × 10^4^ cells resulted in significant growth reduction in KLF4-overexpressed cells compared to empty (without insert)—pCMV3 vector transfected cells seven days after 10 µM Cisplatin treatment (**B**). *: less than 0.05; **: less than 0.01.

**Figure 7 biomedicines-10-02482-f007:**
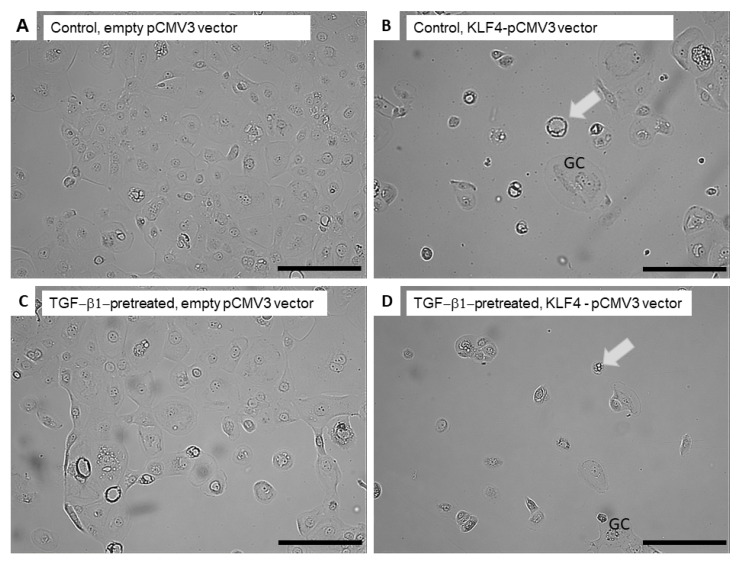
Morphological phase contrast images of SCC-25 cell culture plated at 3 × 10^4^ cells/mL 7 days after the Cisplatin treatment. The cells transfected with empty pCMV3 vector (**A**) show epithelial proliferation with heterogenetic cell morphology both in case of TGF-beta1-pretreated (**C**) and not pretreated cells (**A**). The KLF4-transfected cells are less in number (**B**,**D**), especially if they were TGF-beta1-pretreated (**D**) contain vacuolized cells (arrows) and more nuclear giant cells (GC), which do not proliferate. Scaling bars: 500 µm.

## Data Availability

All data supporting reported results is contained within the article or Appendix A.

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
