# Peer review of "Interplay between Partial EMT and Cisplatin Resistance as the Drivers for Recurrence in HNSCC"

_biomedicines, 2022, doi:10.3390/biomedicines10102482_

Round 1

Reviewer 1 Report

Interesting presentation of an important study concerning highly-relevant Cisplatin resistance. The abstract can be improved by including less methods and more structured presentation of results and conclusions.

Furthermore, other reasons for cisplatin resistance should be discussed according to current literature.

Author Response

Detailed answers to the Reviewer comments

Authors are grateful for the Reviewer for the constructive comments and the contribution to the improvement of the manuscript. Revised text was labeled in red in the manuscript.

Reviewer 1

Interesting presentation of an important study concerning highly-relevant Cisplatin resistance. The abstract can be improved by including less methods and more structured presentation of results and conclusions.

Answer: Authors are grateful for the constructive comments of the Reviewer, and for the valuable suggestions. The abstract session is now rewritten taking the word limits of the Journal also in consideration (page 1, lines 12-25).

Furthermore, other reasons for cisplatin resistance should be discussed according to current literature.

Answer: Required literature is now included and discussed in the discussion part (page 16, lines 551-564).

Unfortunately, the reference insertion and citation management function was blocked in the word document, which the Journal requested to work with during the revision. Reviewer 1 raised comments that required editing and updating the references in the manuscript, which was not allowed and not possible. The new references are now listed extra in the submitted documents. We will work with the Journal to solve this problem.

Reviewer 2 Report

In the study by Ingruber et al the authors investigate EMT and cisplatin resistance in the context of HNSCC.

The study appears robust and encompases both work on clinical samples as well as cell cultures in vitro. The topic covered appears very interesting as well as promising as it might have influence in HNSCC tumor management eventually.

However the presentation of the methods/patients and some of the critical results is not very clear (or inappropriately analyzed) and must be significantly improved.

Particular issues are listed below:

P3L 126 There is not enough information about the primary sites of patient tumor materials. 8th edition of AJCC staging guidelines makes a clear distinction between HPV positive oropharyngeal cancers and oral/HPV negative oropharyngeal cancers. However, in the current study no information about the sites is given despite its potentially great relevance

P6 L279 p16 is used as surrogate marker for HPV which may or may not be correct depending on the particular tumor anatomical site (OK for oropharyngeal, not that OK for others)

P6 L282-292 the paragraph reads more like a figure caption than a part of results. For example, the tumor subsites (lacking above) are now explicitly mentioned just for some particular cases shown on the figure 1. Please decrease the content overlap between the text and actual figure caption P7L301-P8L314

P8 L315 this section is a bit confusing. Apparently 16 regions of primary tumors were assayed as well as corresponding 28 regions in recurrence tumors.  This well corresponds to 44 (16+28) regions described in supplementary table S1B (which is not referred to at this point in the manuscript text). However the S2b table caption implies the regions were assayed from only 7 discrete patients? How did the authors select 16 regions in 7 primary tumors? And conversely how did they decide to assess 28 regions from the recurrent tumors of those 7 patients? The table S1B implies that second sample came from either recurrent, residual or secondary tumors which are probably biologically very distinct and might introduce a lot of uncertainty if dealing only with 7 patients. In either case consider including more detailed explanations within table S1B

P8 L328 aims to compare pre and post therapy images (n=12 vs n=20) and suggests that cells surviving therapy have different expression. However, table S2B suggests that 19 of 44 images came from patients receiving surgical treatment, 9 supportive care and 16 R(CT). It is impossible to determine which therapies were given to the 12+20 subset of images and whether the choice of the subset could affect the data shown

P8 L324 was a paired test used to compare differences in paired samples?

P8 L327 why did the numbers change to n=12 and n=20 for the comparison before and after therapy if all selected 28 regions above are from recurrence tumors?

P9 L344-353 language used seems suboptimal

P10 L 381 inconsistent highlighting of Figure callouts

P10 L382 inconsistent naming of SCC-25 cells

P12 Figure 4 Panels A, B, E-G can be cropped out to reduce the wasted space and or inverted to be better visible. Panels C and D have lower part of barchart cropped out inadvertently. Raw WB images are more appropriate for supplementary material while quantification plots are OK for the main manuscript.

P12 L 398 Figure 4 (and others) description shouldn’t be copy pasted text from the methods

P13 L418 inconsistent naming of UPCI-SCC90 cells

P13 L 444 Figure 5 description shouldn’t be copy pasted text from the methods. L449 suggests 3 expression vectors were used: “control”, “empty” and “KLF4 overexpressing”, but the subsequent text doesn’t identify WB samples where control vector was used.

P13 L431 it is completely unusual to see the Table 1. Consider replacing the Table with some graph. Currently it is not very informative without some mean or median quantification values (with SD, IQR or other measure of spread). Why is the number of experiments sometimes different (11,22,23). If the authors assessed the SLUG OD of sample 1 (serum, empty vector) and OD of sample 3 (5 days TGFB, empty vector) and repeated the measurements 3 times, how do they reach N=12 for the statistical test. (3*2=6)? The presentation of the results is possibly too confusing or the methods not clearly reported, or possibly the authors compared samples which were treated with cisplatin along with those not treated (1&5 vs 3&7 x3) which might not be appropriate to mix or should be explained in methods with strong justification for this decision. Especially problematic would mixing everything for cisplatin.  

P13 L431 the “statistical test” column is completely unnecessary and could be replaced with a table footnote

P13 L465 considering “The normalized ODs of control samples (only serum-free culture condition and empty vector transfection) were set to „1“”  it is unusual to see that “not treated” column of Fig S3A is not =1 as this should be the same sample in triplicate (empty vector (panels A-D), not treated by TGFB (=serum)

P13 L 431-435 there is no Supplementary figure S2 with panels mentioned herein

P13 L444 Figure 5. Instead of presenting western blot image of a single experiment, consider presenting better organized quantification graphs (like in Fig 4H) in the manuscript and less informative western blots in supplement. The Supplementary figure 3 doesn’t try to link the graphs shown to the extensive sample identification numbers with different combination of treatments or vectors (previously named 1, 2, 3, …8).

P14 P470 this paragraph is mixing too many separate concepts. At least confluence measurements (why not the usual viability/proliferation assay?) should  be in a new paragraph if not separate subheading . In either case the order of experiments should more closely follow the materials section.

P14 L472 the supplementary document on page 10 should either explain the different sequence highlights (colours, font and underlines) or not present those sequence highlights. Some reference for the binding motif would be helpful.

P15 L477 The methods (P5L210-216) describing the confluence measurements do not clearly explain the figure . The methods suggest 7 timepoints: “Cell growth was evaluated…” “end of the TGF-beta1-treatment”, “on day 10”, “at the end of the vector transfection”, and “2”, “4”, “7” and “9 days after replating following the Cisplatin-treatment. The figure 6 contains only the 4 last timepoints

P15 L490 the description of Figure 6 is unclear. It is difficult to understand what part of the text corresponds to which subpanel. Methods (P5L210-216)  do not mention 4 hours of cisplatint treatment while the figure legend suggests that the majority of figure (panels A and B) present data of 4 hour cisplatin treatment and that only panel C is in line with methods. Panel B seems to present only the 7 days datapoint of panel A with different error bars and might be erroneous and or obsolete. Panels B and C despite similarity have different Y axis maximums. Why doesn’t panel A show the 0 datapoint to show that plated cell density was the same at the start of the experiment

P15 L497 the paragraph is again very difficult to read since it is mostly a single sentence.

P17L539-553  Large parts of the discussion do not attempt to put the current results in the context of work by other authors but only rephrase results. Or conversely just repeat results of other studies without directly linking them to the current work (i.e. P17 L555-562). The authors should aim to improve the discussion in lines with the paragraph on cisplatin (P17 L563-574) which quite appropriately puts the current results in context.

Author Response

Please find attached in the pdf file the detailed answers to Reviewer 2 comments.

kind regards

Round 2

Reviewer 2 Report

The revised manuscript by Ingruber et al is a much improved version addressing all issues raised.

No further review is needed but the authors should fix few typos and consider some very small improvements that would benefit future readers.

P3 L125 Typo “Supplementary Table. S1”  extra fullstop.

Regarding issue 2 dealing with p16. The response to reviewers letter addresses the question sufficiently, but i strongly suggest that HPV DNA of p16+ cases is acknowledged in supplementary table 1 (ie. with a footnote) for all readers and not only reviewers. Not everyone knows Innsbruck hospital protocols and HPV DNA analysis is still not standard practice in all research.

Supplementary table S1 B. Consider rearranging the last two rows so that the “primary tumor” treatment modality is given first since it is more relevant to the recurrence outcome. While not really necessary the authors might consider separating the Primary/recurrence field as columns and present different parameters in primary tumor, recurrent tumor and keep the total as currently shown. In this way there would be no ambiguity that Primary/Recurrence subgroups are somehow biased.

P12 L399 typo  “Supplementary Figure S4 B,)” extra comma

Author Response

Please find attached the file "Response to reviewer2_minor revision".

Best,

Julia Ingruber
